# DB-KSVD: Scalable Alternating Optimization for Disentangling High-Dimensional Embedding Spaces

Romeo Valentin [1]   Sydney M. Katz [1]   Vincent Vanhoucke [2]   Mykel J. Kochenderfer [1]

## Abstract

Dictionary learning has recently emerged as a promising approach for mechanistic interpretability of large transformer models. Disentangling high-dimensional transformer embeddings requires algorithms that scale to high-dimensional data with large sample sizes. Recent work has explored sparse autoencoders (SAEs) for this problem. However, SAEs use a simple linear encoder to solve the sparse encoding subproblem, which is known to be NP-hard. It is therefore interesting to understand whether this approach is sufficient to find good solutions to the dictionary learning problem or if a more sophisticated algorithm could find better solutions. In this work, we propose Double-Batch KSVD (DB-KSVD), a scalable dictionary learning algorithm that adapts the classic KSVD algorithm. DB-KSVD is informed by the rich theoretical foundations of KSVD but scales to datasets with millions of samples and thousands of dimensions. We demonstrate the efficacy of DB-KSVD by disentangling text embeddings of the Gemma-2-2B and Pythia-160M models and evaluating on six metrics from the SAEBench benchmark, where we achieve competitive results when compared to established approaches based on SAEs. We further show similar results when disentangling image embeddings obtained from the DINOv2-S and DINOv2-B models, solidifying our findings. By matching SAE performance with an entirely different optimization approach, our results suggest that (i) SAEs do find strong solutions to the dictionary learning problem and (ii) traditional optimization approaches can be scaled to the required problem sizes. We make an implementation of DB-KSVD available at `https://github.com/romeov/ksvd.jl`.

## 1. Introduction

Large transformer models have enabled a wide range of applications in natural language processing (Vaswani et al., 2017), computer vision (Dosovitskiy et al., 2021), and numerous other fields (Dalla-Torre et al., 2025; Wen et al., 2023; Brohan et al., 2023). To deploy these models in high-stakes applications, it is desirable to understand their internal representations to validate their reasoning and detect potential biases in their world models. However, interpreting these internal representations remains a significant challenge because they cannot easily be decomposed into monosemantic and interpretable features. Instead, it has been hypothesized that these internal representations live in a highly-entangled vector space, in which monosemantic features are superimposed (Elhage et al., 2022).

Following this hypothesis, in this work, we focus on the disentanglement of these internal representations with the goal of mapping entangled latent embeddings to their corresponding monosemantic features. More specifically, we assume that any latent embedding sample $y$ can be generated by a sparse linear combination of columns of a fixed and unknown dictionary $D$, i.e., $y = Dx$, where $x$ is a sparse vector. In this formulation, the dictionary $D$ comprises monosemantic feature representations and serves as an overcomplete basis for the embedding space. The sparse vector $x$ encodes the presence of each monosemantic feature in the embedding. For further intuition about these concepts, we refer to Section A for a visualization.

The goal of dictionary learning is then to find this unknown dictionary $D$ given only a finite set of samples $\{y_i\}_{i \in 1..n}$. Finding $D$ enables the decomposition of an embedding into its monosemantic feature activations $x$, and possibly allows for interpreting each monosemantic feature representation. Beyond the disentangling of transformer embeddings, dictionary learning has also had various other applications such as signal recovery and image compression (Tošić & Frossard, 2011). Although this problem is known to be NP-hard (Razaviyayn et al., 2014), a class of dictionary learning algorithms based on alternating optimization (AO) has been proposed with convergence guarantees under some assumptions (Agarwal et al., 2016). However, these guarantees require strong assumptions that typically do not hold

---

[1] Stanford University [2] Waymo. Correspondence to: Romeo Valentin <romeov@stanford.edu>.

*Proceedings of the 43rd International Conference on Machine Learning*, Seoul, South Korea. PMLR 306, 2026. Copyright 2026 by the author(s).

in practice. Nevertheless, proposed algorithms for approximate solutions (Aharon et al., 2006; Rubinstein et al., 2008; Irofti, 2014; Lillelund et al., 2022; Spielman et al., 2012; Patel & Chellappa, 2011; Zhang et al., 2015) have been widely studied in the context of small to medium scale problems, but they do not immediately scale to the large, high-dimensional datasets that we encounter in the context of transformer models.

To apply dictionary learning to the disentanglement of transformer embeddings, we therefore need to scale dictionary learning algorithms to large problem sizes. Recent work has explored the use of sparse autoencoders (SAEs) as a scalable alternative to established dictionary learning algorithms (Bricken et al., 2023; Templeton et al., 2024; Cunningham et al., 2023; Gao et al., 2024; Bussmann et al., 2025). SAEs typically comprise a simple linear encoder and decoder layer and are trained using gradient descent. However, it is not clear whether the solutions found by SAEs approach the global optimum or whether previously proposed AO approaches may perform better. For this reason, we introduce DB-KSVD, an adaptation of the classic AO-based KSVD algorithm (Aharon et al., 2006) that can be applied to datasets consisting of millions of samples with thousands of features.

Our contributions are as follows: **(i)** We present **DB-KSVD, a scalable adaptation of the KSVD algorithm** that can be applied to large-scale datasets and scales well with CPU and GPU availability. **(ii)** We examine **theoretical aspects of the tractability of the dictionary learning problem** in terms of sampling complexity, sparsity, and the number of identifiable features and link this analysis to design considerations for our algorithm. **(iii)** We adopt a key idea from the recent work on SAEs, the **Matryoshka encoding structure** (Bussmann et al., 2025), to DB-KSVD and show performance improvements in some metrics. **(iv)** We apply DB-KSVD to disentangle embeddings of the Gemma-2-2B model (Mesnard et al., 2024) and **show competitive performance of our learned dictionaries when compared to established SAE-based approaches** on six metrics of the SAEBench benchmark. We further solidify these findings with similar results when using the Pythia-160M (Biderman et al., 2023) and DINOv2 (Oquab et al., 2024) models, where the latter is applied to a vision task as opposed to a language task.

## 2. Preliminaries

The goal of dictionary learning, also known as sparse coding, is to model a set of data samples as linear combinations of a few elementary signals. Formally, given a data matrix $Y = [y_1, \ldots, y_n] \in \mathbb{R}^{d \times n}$, we want to find a dictionary $D = [d_1, \ldots, d_m] \in \mathbb{R}^{d \times m}$ where $\|d_i\|_2 = 1$ and a sparse coefficient matrix $X = [x_1, \ldots, x_n] \in \mathbb{R}^{m \times n}$ where

$\|x_i\|_0 \leq k$ such that

$$Y \approx DX \tag{1}$$

and $n \gg m > d$. Specifically, we consider the optimization problem

$$\min_{D,X} \|Y - DX\|_F^2 \quad \text{s.t.} \quad \|d_i\|_2 = 1, \quad \|x_i\|_0 \leq k. \tag{2}$$

In the context of disentangling transformer embeddings, the data matrix $Y$ consists of the embeddings of the transformer model, and we are trying to find an overcomplete dictionary $D$ with columns that represent monosemantic features of the embeddings.

One family of algorithms solves the optimization problem in Eq. (2) using AO of two subproblems: (i) finding $X$ for a fixed $D$ and (ii) improving $D$ for a fixed structure of $X$. We refer to these two steps as the sparse encoding step and the dictionary update step, respectively. In this work, we adapt the KSVD algorithm (Aharon et al., 2006), which uses these AO steps. The remainder of this section provides a brief overview of the KSVD algorithm and its two main components.

**Sparse Encoding.** The sparse encoding step aims to find a sparse vector $x_i$ for each data sample $y_i$ given a fixed dictionary $D$. This problem is NP-hard (Natarajan, 1995) due to the combinatorial nature of the sparsity constraint, and we must rely on approximate solutions. A number of algorithms have been proposed, including Matching Pursuit (MP) (Mallat & Zhang, 1993), Orthogonal Matching Pursuit (OMP) (Pati et al., 1993), LASSO regression (Tibshirani, 1996), and solving a mixed-integer program (Liu et al., 2019). In this work, motivated by computational efficiency, we adapt the MP algorithm, which greedily selects dictionary elements $d_j$ and repeatedly updates a residual $r$ as

$$r \leftarrow r - \langle r, d_j \rangle d_j \quad \text{where } j = \arg\max_{j'} \langle r, d_{j'} \rangle \tag{3}$$

until the sparsity constraint becomes active. Using the computational modifications introduced in Section 4.1, we can typically solve the sparse coding problem in less than one millisecond per sample, even for large dictionaries.

**Dictionary Update.** The dictionary update step updates each dictionary element $d_j$ by considering a subset of the error matrix that contains the residuals of the samples that use the $j$th dictionary element. Formally, we update each column $d_j$ using the error matrix

$$E_{\Omega_j} = Y_{\Omega_j} - DX_{\Omega_j} \tag{4}$$

where $\Omega_j$ denotes the set of column indices of the samples that use the $j$th dictionary element. We then replace $d_j$ with the first left singular vector of the error matrix $E_{\Omega_j} - d_j X_{j,\Omega}$,

i.e., the error matrix without the contribution of the $j$th dictionary element. For the KSVD algorithm, we also update the values of the nonzero elements in the corresponding row $X_{j,\Omega}$ using the first right singular vector and singular value. Using this process, all dictionary elements are updated sequentially. Although the dictionary update step typically dominates the runtime of the KSVD algorithm, the computational modifications presented in Section 4.1 allow us to significantly decrease the runtime to the point where it roughly matches that of the sparse encoding step.

# 3. Theoretical Aspects of Sparse Encoding and Dictionary Learning

As introduced in Section 1, we motivate the model $Y \approx DX$ for transformer embeddings with the superposition hypothesis. However, even if this model is accurate, it is not clear whether the dictionary learning problem is identifiable, i.e., whether there can exist an algorithm that can actually recover the dictionary $D$ and sparse assignments $X$ solely from data samples $Y$. For this reason, it is useful to understand our problem in the context of the well-established theory of the dictionary learning problem and the sparse encoding subproblem. We use this theory to motivate the need for a scalable adaptation of the KSVD algorithm and to inform our algorithmic and experimental design choices.

### 3.1. The Ill-Posed Nature of Dictionary Learning

Without sparsity constraints on $X$, the dictionary learning problem from Eq. (1) is ill-posed in the sense that an infinite number of solutions $(D, X)$ exist. This non-identifiability issue is a common challenge in unsupervised learning. For example, Locatello et al. (2020) showed that finding monosemantic disentangled representations from observational data without appropriate inductive bias is impossible. However, for many types of data such as images or natural language, we can impose additional constraints on the dictionary learning problem. In particular, we will assume that each data sample is only composed of a few monosemantic features, which motivates a sparsity constraint on the coefficient matrix $X$. This constraint transforms the problem from an underconstrained and non-identifiable problem to a sparse dictionary learning problem, which is potentially identifiable.

However, even with the sparsity constraint, the dictionary learning problem is not guaranteed to be identifiable. Instead, the identifiability further depends on problem parameters such as the dimensionality of the embeddings $d$, the sample size $n$, the number of dictionary elements $m$, the level of sparsity $k$, the "level of orthogonality" (incoherence) of the dictionary elements $d_i$, and the magnitude of noise or unmodeled terms. To understand the impact of these parameters in the context of our problem, we turn to

the theoretical aspects of the dictionary learning and sparse encoding problems. For example, for low levels of sparsity (high $k$), just solving the sparse encoding subproblem of finding $x$ given $y$ and a fixed $D$ is already difficult or infeasible, regardless of the algorithm used. Furthermore, even if the sparse encoding problem is feasible for a fixed $D$, the simultaneous identification of $D$ and $X$ is even more challenging. The remainder of this section aims to build intuition for how the difficulty of the dictionary learning problem depends on the problem parameters.

### 3.2. The Identifiability of Sparse Encoding

The identifiability of the sparse encoding subproblem is fundamentally a function of the sparsity $k$ and the incoherence of $D$. Intuitively, low sparsity admits exponentially many more possible combinations of the columns of $D$. Furthermore, when the columns of $D$ are near parallel, or "highly coherent", it is difficult to determine which columns were used to generate a given data sample $y$. Formally, the coherence of $D$ is defined as $\mu(D) = \max_{i \neq j} |\langle d_i, d_j \rangle|$ and provides a measure of the orthogonality of the dictionary elements. We can relate the coherence of $D$ to the maximum value for $k$ such that the sparse vector $x$ is unique and can be recovered through algorithms such as OMP or $L_1$-minimization. Specifically, the condition

$$k < \frac{1}{2}(1 + \mu^{-1}(D)) \tag{5}$$

is a sufficient condition (Tropp & Gilbert, 2007, Thm. B) for recoverability. For $k$ to be as large as possible, we want $D$ to have minimal coherence, i.e., to have maximally orthogonal columns. We find that KSVD, by default, produces highly-coherent dictionaries, which motivates the extension described in Section 4.2. We note that coherence can be a "blunt instrument" because it only considers the maximum coherence, and the related Restricted Isometric Property (RIP) can be used as a more nuanced metric (Candès et al., 2006; Candès & Wakin, 2008; Foucart & Rauhut, 2013). We look forward to future research exploring RIP in the context of transformer embeddings.

We can also use Eq. (5) to relate the number of recoverable features to the dimensionality of the samples. The coherence of maximally incoherent dictionaries is limited by the Welch bound (Welch, 1974) with $\mu(D) \geq \sqrt{(m-d)/(d(m-1))}$. If the number of dictionary elements $m$ is a similar order of magnitude compared to the embedding dimension $d$, we can approximate the bound as $\mu(D) \geq \sqrt{1/d}$. Plugging this result into Eq. (5), we find that in the optimal case of maximally incoherent dictionaries the number of guaranteed identifiable elements scales with $\sqrt{d}$. This result indicates that the dimensionality $d$ of the data must be sufficiently larger than the number of monosemantic features $k$ in each sample.

### 3.3. Information-Theoretic Limits for Sample Complexity of Dictionary Learning

In the previous section, we discussed the challenge of finding $X$ given $Y$ and $D$. However, even when the sparse encoding subproblems for each $y$ are identifiable, simultaneously identifying $D$ and $X$ is even more challenging and generally requires a large number of samples. This challenge is exacerbated further by the fact that $Y = DX$ is an imperfect model, and we instead have an unmodeled term $\epsilon$ such that $Y = DX + \epsilon$. Jung et al. (2016) take an information-theoretic approach to characterize the required number of samples for dictionary identifiability in this setting. They assume a Gaussian distribution for both the nonzero coefficients in $X$ and the unmodeled term $\epsilon$ with variances $\sigma_X^2$ and $\sigma_\epsilon^2$, respectively. The authors establish a lower bound on the required number of samples that is proportional to $m^2$ and inversely proportional to the signal-to-noise ratio SNR $= \sigma_X^2/\sigma_\epsilon^2$ (Jung et al., 2016, Eq. 21). Intuitively, this relationship indicates that (i) learning additional dictionary elements requires a quadratic number of additional samples and (ii) we can only identify nonzero elements if their signal is sufficiently large compared to the unmodeled term or "noise" $\epsilon$.

### 3.4. Practical Considerations

Returning to our application of disentangling large transformer embeddings such as the embeddings from the Gemma-2-2B model (Mesnard et al., 2024), we propose the following practical considerations: **(i)** The large embedding dimension of transformer models ($d = 2304$ for Gemma-2-2B) benefits the number of recoverable monosemantic features $k$ per sample. Conversely, if this method is applied to embeddings with a smaller dimension, the number of recoverable features may be smaller. **(ii)** The incoherence of the learned dictionary $D$ is an important characteristic of the feasibility of the dictionary learning problem, and it can be used as a diagnostic tool to indicate dictionary performance on downstream metrics. **(iii)** Although the superposition hypothesis suggests a larger number of dictionary elements $m$ than the data dimension $d$, we cannot arbitrarily increase $m$ without simultaneously scaling the number of data samples $n$ (quadratically under a Gaussian sparse signal assumption).

## 4. Double-Batch KSVD

In the previous section, we highlighted the challenges of the dictionary learning problem, and it may seem a miracle that we can find satisfactory results at all. Fortunately, in the case of transformer embeddings, the identifiability of the dictionary learning problem benefits from the large embedding dimension and sample sizes. However, exploiting these benefits requires scaling to large problem sizes, which mo-

tivates modifications to the classic KSVD algorithm. This section outlines the main components of the DB-KSVD and Matryoshka DB-KSVD algorithms that allow us to scale to millions of data samples and thousands of dimensions, reducing the required runtime for DB-KSVD from weeks to minutes.

### 4.1. Scaling KSVD to Millions of Samples

Previous efforts have proposed a variety of algorithmic modifications to improve the performance of the KSVD algorithm on single-core (Rubinstein et al., 2008), multi-core (Sukkari et al., 2017), and hardware-accelerated systems (Irofti, 2014; He et al., 2016). However, these implementations have only been scaled to datasets with hundreds of dimensions and tens of thousands of samples, multiple orders of magnitude smaller than our requirements. In this section, we discuss algorithmic modifications to the classic KSVD algorithm that make it possible to scale it to thousands of dimensions and millions of samples. Specifically, we make modifications that allow us to reduce the computations needed in each step of the algorithm and map the problem to a scalable number of CPU workers. We also discuss how hardware acceleration can optionally be used to offload the most expensive operations, specifically those that scale with the number of samples $n$. Finally, we show how batching can be used to scale to even larger sample sizes beyond the limits of the working memory in a similar manner to how mini-batching is done for many machine learning workflows.

**Parallel Matching Pursuit.** The sparse encoding step of the KSVD algorithm initially appears to be "embarrassingly parallel" in the samples $y_i$ such that it can be easily scaled to a large number of workers. However, when encoding a large number of samples with the same dictionary $D$, the performance can be enhanced significantly. From Eq. (3), we can see that $\langle r, d_j \rangle$ are elements of $D^\top r$, which we can store. Instead of recomputing $D^\top r$ after every update, we can notice the recurrence relation $D^\top r^{(t+1)} = D^\top r^{(t)} - \langle r, d_j \rangle (D^\top d_j)$. Further, we can fully avoid computing any matrix-vector products during the sparse encoding iterations by precomputing $D^\top D$, which is constant across all iterations and all samples $y_i$. For a more detailed explanation, we refer to Section 3.3 of (Davis et al., 1997) where similar optimizations have been proposed. We can also initialize the product vector $D^\top r^{(0)}$ for each sample $y_i$ by precomputing the matrix $D^\top Y$ and initializing each product vector with the corresponding column. With $D^\top D$ and $D^\top Y$ precomputed, each of the $k$ iterations for each sample reduces to (i) an $\arg\max$ operation over the precomputed product vector $D^\top r^{(t)}$, (ii) indexing into $D^\top r^{(t)}$ to retrieve $\langle r, d_j \rangle$, and (iii) updating the product vector through a simple vector addition.

This optimized approach significantly shifts the computational burden. The precomputation of $D^\top D$ and $D^\top Y$ requires $O(dm^2)$ and $O(dmn)$ operations, respectively. In contrast, the subsequent $k$ sparse encoding iterations for each of the $n$ samples involve approximately $O((m+d)kn)$ operations. Since $k \ll d < m$, the cost of precomputing $D^\top D$ and $D^\top Y$ dominates the overall runtime of the entire sparse encoding problem when using MP. When heterogeneous compute is available, we can further speed up the computation by offloading the two matrix products to hardware accelerators such as GPUs. In practice, we find that this approach can be scaled efficiently to both high-CPU and heterogeneous CPU-GPU machines.

**Inner Batching: Batched and Accelerated Dictionary Updates.** The dictionary update step of the original KSVD algorithm has an inherently sequential nature. Specifically, in Eq. (4), when updating the $j$th dictionary element, we require the modified $D$ and $X$ matrices resulting from the previous dictionary element update. One way to circumvent this sequential nature and parallelize the dictionary update step is to use $D$ and $X$ of the previous KSVD iteration for all dictionary elements, rather than from the most recent update. However, this approach may deteriorate the convergence properties of the algorithm (Aharon et al., 2006, Sec. IV.C).

Instead, we propose to partition the $m$ dictionary update tasks into shuffled batches of size $w$, where $w$ is the number of available workers, e.g., CPU threads. Next, each worker performs one dictionary update on a local copy of the most recent dictionary $D$ and sparse assignment matrix $X$. After all workers simultaneously perform their update, the updated column in $D$ and row in $X$ is synchronized with all other workers in an all-to-all fashion. Then, the next batch of $w$ dictionary update tasks is considered and the process is repeated until all dictionary elements have been updated. Empirically, this batched scheme does not significantly deteriorate convergence speed while allowing highly efficient scaling of the dictionary update tasks to many CPU workers; we discuss convergence considerations in more detail below.

Next, we turn to the effect of the number of samples $n$ on the computational cost of the dictionary update step. If we assume the nonzero elements in $X$ are evenly distributed, the error matrix $E_{\Omega_j}$ will have approximately $nk/m$ columns. As we increase $n$, computing the maximum singular vector of $E_{\Omega_j}$ can become a computational bottleneck. We address this bottleneck in three steps. First, we note that we can avoid computing the full singular value decomposition of $E_{\Omega_j}$ by using an iterative algorithm to compute only the maximum singular vector. Second, the singular vectors of $E_{\Omega_j}$ can be computed from the eigenvectors of either $E_{\Omega_j}^\top E_{\Omega_j}$ or $E_{\Omega_j} E_{\Omega_j}^\top$. For large $n$, we can therefore compute the maximum eigenvector of the symmetric matrix $E_{\Omega_j} E_{\Omega_j}^\top \in \mathbb{R}^{d \times d}$ with an iterative algorithm. We find that

the Lanczos algorithm (Lanczos, 1950) performs best for this problem.

For each of the $m$ dictionary elements, the computational complexity of the dictionary update is dominated by forming $E_{\Omega_j} E_{\Omega_j}^\top$ which requires amortized $O(d^2 kn/m)$ operations. Finding the largest eigenvalue of this matrix requires $O(d^2)$ operations, where we assume a fixed maximum number of iterations. We point out that the computational complexity of the eigenvalue problem no longer depends on the number of samples $n$. Further, computing $E_{\Omega_j} E_{\Omega_j}^\top$ can also be offloaded to hardware acceleration if available, providing significant additional speedup. However, to recover the singular vector from the computed eigenvector, a single multiplication with $E_{\Omega_j}$ is still necessary, so we cannot offload the entire dependence on $n$. Nonetheless, this optimization enables us to significantly reduce this dependence, allowing us to scale to problem sizes with millions of samples and to leverage both high-CPU and heterogeneous CPU-GPU machines.

**Outer Batching: KSVD for Out-of-Core Problem Sizes.** The algorithmic modifications to the sparse encoding and dictionary update steps, combined with careful reduction of memory allocations, memory movement, and threading overhead, allow us to efficiently process up to one million samples at a time. However, for problems with more samples, the KSVD update step as written so far will exceed memory capacity. To remedy this problem, instead of using all available data at each DB-KSVD iteration, we propose batching the data in a similar manner to mini-batching for gradient descent-based training. In particular, at each step of DB-KSVD, we execute the sparse encoding and dictionary update steps on only a subset of the data samples. We find that this batching step is also beneficial for problem sizes that fit into the working memory. For a fixed time budget, the increased number of DB-KSVD iterations due to smaller batches outweighs the benefit of using the maximum number of samples in each step, provided that the batch size is large enough (typically one to two orders of magnitude larger than the data dimension $d$).

**Computational Complexity.** Combining the sparse encoding and dictionary update step, replacing the number of samples $n$ with the batch size $n_b$, and assuming a fixed number of DB-KSVD iterations, we can derive an overall computational complexity of $O(dm(m + n_b) + d^2(kn_b + m))$. If we further choose $m$ as a fixed multiple of $d$ (a common choice), similarly choose $n_b$ to be one or two orders of magnitude larger than $d$, and recall $k \ll d$, we can simplify the computational complexity to $O(d^3)$. We provide additional details on complexity and memory requirements in Section C.

**Convergence Considerations.** We note that due to the batching and shuffling, DB-KSVD is not exactly mathe-

matically equivalent to the original KSVD algorithm, and we therefore can't trivially claim convergence guarantees stated for the KSVD algorithm in the literature (Aharon et al., 2006; Bao et al., 2015; Agarwal et al., 2016; Chatterji & Bartlett, 2019; Ruetz & Schnass, 2026). However, the exact guarantees are unlikely to hold for our setting in the first place: they typically rely on assumptions about the initialization which we can not make, whether for KSVD or DB-KSVD. Notwithstanding, we argue that the structural benefit of alternating minimization, i.e., the closed-form updates over each block conditioned on the other, continues to hold. Empirically we find that the simultaneous batched updates with shuffled order across outer batches do not significantly deteriorate convergence speed.

### 4.2. Imposing Inductive Bias with Matryoshka Structuring

As outlined in Section 3.2, incoherence is an important property of the learned dictionaries. However, as we present in Section 5.1, the dictionaries learned by DB-KSVD tend to have high coherence, especially for small values of $k$. We therefore propose additional structure as a regularization technique to encourage more incoherent dictionaries. A promising approach for imposing additional structure is the Matryoshka structuring, which was recently proposed in the context of SAEs (Bussmann et al., 2025) and representation learning (Kusupati et al., 2022). The Matryoshka structuring introduces a hierarchical ordering of dictionary elements that aims to encourage different layers of semantic detail.

We implement the Matryoshka structuring as follows. Following (Bussmann et al., 2025), we partition the dictionary elements into groups of exponentially increasing size. Each group is assigned an equal share of the total budget of $k$ nonzero elements. At each iteration of the KSVD algorithm, instead of doing a single sparse encoding and dictionary update step, we begin by performing these steps using the first group of dictionary elements and assignments only, which we refer to as $D^{(1)}$ and $X^{(1)}$. We then compute the residual matrix $E^{(1)} = Y - D^{(1)}X^{(1)}$ and fit the next group of dictionary elements to this residual matrix $E^{(1)}$. We repeat this process for all groups. While we perform the sparse encoding step on all subgroups separately during training, we find that a single sparse encoding step on the full dictionary is sufficient at evaluation time.

## 5. Experiments

To evaluate the efficacy of DB-KSVD for disentangling transformer embeddings, we apply it to embeddings from the Gemma-2-2B and Pythia-160M language models. To evaluate the quality of the found dictionaries, we follow the SAEBench benchmark (Karvonen et al., 2025) and compute six metrics on the resulting dictionaries comprising loss

recovered, autointerpretability, absorption, sparse probing, spurious correlation removal, and attribute-value entanglement. We then study the coherence of the learned dictionaries and analyze the impact of induced bias from Matryoshka structuring. To show the scalability of DB-KSVD we report timing results of our CPU-only and hardware accelerated implementations and compare against an implementation of the KSVD algorithm as introduced by (Aharon et al., 2006). To further validate these results beyond the language domain, we also apply and evaluate DB-KSVD on embeddings from DINOv2-S/-B models for a vision task.

### 5.1. SAEBench Results

We evaluate the performance of our trained dictionaries from the DB-KSVD algorithm and the Matryoshka adaptation on the SAEBench benchmark by constructing dictionaries from 2.6 million embeddings of the Gemma-2-2B ($d = 2304$) and Pythia-160M ($d = 768$) models evaluated on the Pile Uncopyrighted dataset (Gao et al., 2020). We learn dictionaries of size $m \in \{4096, 16384\}$ with sparsity $k \in \{20, 40, 80\}$ by running 40 iterations of batch size $n = 2^{16}$ (specific training details are summarized in Section B). These parameters are chosen because we notice no further performance improvements in our proxy metrics for more data or iterations (see Section D).

We present the results for Gemma-2-2B with $m = 4096$ in Fig. 1. The dictionaries learned using DB-KSVD outperform the Standard ReLU SAE approach (Cunningham et al., 2023; Bricken et al., 2023) in all metrics except autointerpretability and one instance of sparse probing. Moreover, the results are also generally competitive with all other SAE variants including the MatryoshkaBatchTopK SAE (Bussmann et al., 2025), except in the case of autointerpretability. Results for $m = 16384$ as well as for Pythia-160M are provided in Section D and show similar trends. The fact that two completely different optimization approaches (DB-KSVD and SAEs) achieve similar performance results may indicate that we are close to the theoretical limits given the problem size.

We hypothesize that the differences in autointerpretability are related to the coherence of the learned dictionaries, which we analyze in the next section. We also notice that the Matryoshka structuring approach improves autointerpretability and absorption performance; however, it slightly degrades the performance on the loss recovered and SCR metrics. These trends are similar to the trends observed when using the SAEs with Matryoshka structuring, hinting at a more fundamental phenomenon.

### 5.2. Coherence Metrics

To further understand the results from the previous section, we recall from Section 3 that dictionary coherence is an

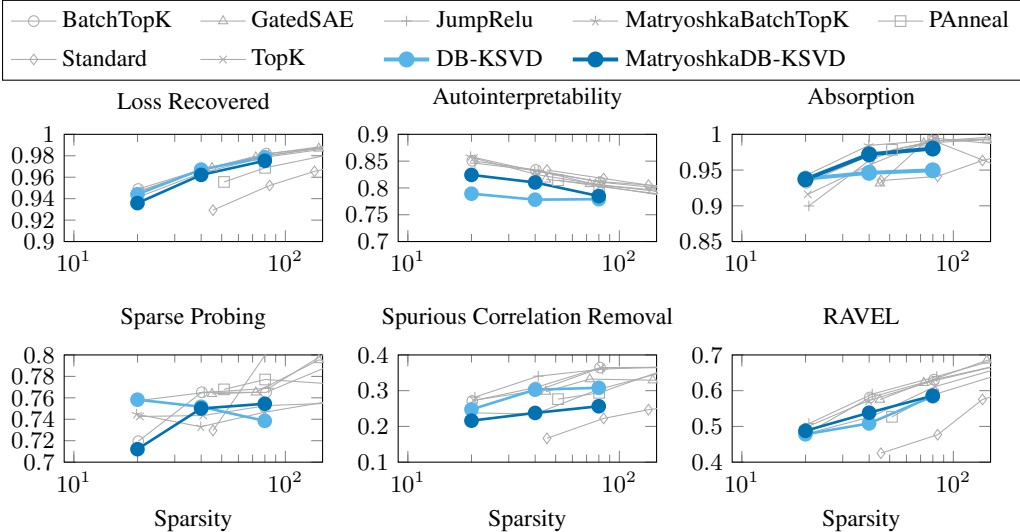

*Figure 1.* Results (higher is better) of our DB-KSVD algorithm and Matryoshka adaptation on the SAEBench benchmark applied to Gemma-2-2B embeddings with 4096 dictionary elements.

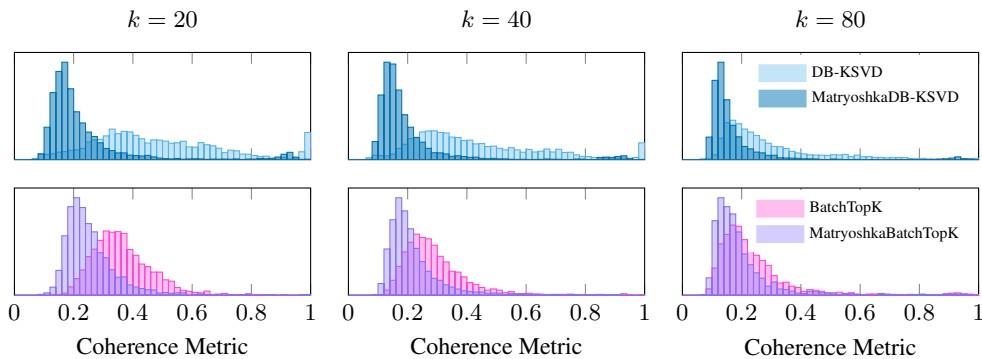

*Figure 2.* Histograms of element-wise coherence metrics for different sparsities. We hypothesize that lower is better. The dictionaries are constructed using the Gemma-2-2B embeddings and comprise $m = 4096$ dictionary elements.

important property for the well-posedness of the sparse encoding subproblem, with high coherence resulting in an infeasible problem. We study the coherence of the learned dictionaries by computing $\max_{\ell \neq j} \langle d_j, d_\ell \rangle$ for each dictionary element $d_j$ and show the results for varying sparsity levels $k$ in Fig. 2. We also compare to SAE dictionaries constructed by (Karvonen et al., 2025) for the same problem. We find that the learned dictionaries for DB-KSVD are relatively coherent, especially for smaller values of $k$. Additionally, there is a notable spike in dictionary elements that are almost perfectly aligned, i.e., $\langle d_j, d_\ell \rangle \approx 1$. This result motivates the extension introduced in Section 4.2 to encourage more incoherent dictionaries. When using the Matryoshka structuring approach, we indeed observe a significant decrease in the coherence of the learned dictionaries. We observe a similar trend when the Matryoshka structuring is introduced to the SAE approach. Furthermore, we hypothesize that the improvement in dictionary

incoherence relates to the improved autointerpretability of MatryoshkaDB-KSVD over DB-KSVD. However, it is unclear whether this result is due to improved sparse encoding performance or the dictionaries themselves.

### 5.3. Timing results

Table 1 shows the runtime results for one DB-KSVD iteration for different problem sizes and compute types. The CPU only results are generated on a Google Cloud Platform n2-standard-128 machine, and the hardware accelerated (CPU+GPU) results are generated on a n1-standard-96 machine with an NVIDIA T4 GPU. DB-KSVD is up to 10 000 times faster than the baseline implementation of KSVD, which still has access to all CPU cores for matrix operations. This result highlights the importance of our algorithmic modifications for the scalability of the DB-KSVD algorithm. In practical terms, for one full run on the 2.6 million embeddings with 4096 dictionary elements and $k = 20$,

| | | CPU | CPU+GPU | Baseline (est.) |
|---|---|---|---|---|
| $m =$ 4096 | $k = 20$ | 5.47 | 11.43 | $6.5 \times 10^4$ |
| | $k = 40$ | 10.86 | 24.03 | $1.0 \times 10^5$ |
| | $k = 80$ | 337.63 | 486.43 | $1.6 \times 10^5$ |
| $m =$ 16384 | $k = 20$ | 17.12 | 23.23 | $7.4 \times 10^4$ |
| | $k = 40$ | 21.51 | 31.77 | $1.3 \times 10^5$ |
| | $k = 80$ | 41.95 | 72.79 | $2.6 \times 10^5$ |

*Table 1.* Runtime in seconds for a single batch of $n = 2^{16} = 65\,536$ Gemma-2-2B embeddings. We report the fastest of 5 trials, except baseline, which is estimated based on timing results from a smaller problem. All CPU only and CPU+GPU trials are within a 10 % margin.

| Method | Recon | SpProb | AutoInterp |
|---|---|---|---|
| SAE | 0.794 | **0.932** | 0.797 |
| DB-KSVD | **0.817** | 0.893 | **0.802** |

*Table 2.* DINOv2-B evaluation at sparsity $k = 32$. Sparse probing uses $k_{\mathrm{sp}} = 1$. Full results are provided in Section D.5.

the DB-KSVD algorithm takes approximately 8 minutes to converge, while the baseline would take over 30 days.

For this problem size, the hardware acceleration does not improve the runtime. Results for larger problem sizes are presented in Section D. For $m = 4096$, the runtime drastically increases when increasing $k$ from 40 to 80 for both compute types. However, this trend does not appear for $m = 16\,384$. We explain this increase by noticing that for a fixed $k$, a smaller $m$ will result in a more dense $X$ and therefore a wider $E_{\Omega_j}$ matrix. This increased width shifts the computational bottleneck, resulting in a larger runtime.

### 5.4. Vision Model Evaluation

To validate whether our findings extend beyond the text domain, we evaluate DB-KSVD on DINOv2-S ($d = 384$) and DINOv2-B ($d = 768$) image embeddings. We extract CLS token embeddings from the approximately 1.3 million image samples of ImageNet-1k (Russakovsky et al., 2015) and learn dictionaries of size $m = 4096$. Since no equivalent to SAEBench exists for the vision domain, we adapt three core metrics from SAEBench and evaluate variance explained, sparse probing accuracy, and VLM-based autointerpretability. For autointerpretability, we select images with the highest activations for a feature index and use GPT-4o-mini to find their commonalities, generating a natural language explanation. We then score the explanation by computing the Spearman correlation between CLIP similarities of (explanation, image) tuples with the class activation label. Further experimental details are provided in Section B.2.

A summary of the results is presented in Table 2 for DINOv2-B with $k = 32$, and further results are provided in Section D.5. We find that similarly to the SAEBench results, DB-KSVD has comparable performance to SAEs across all metrics and sizes. In particular, DB-KSVD consistently slightly outperforms the SAE on variance explained and autointerpretability, with the gap being significantly larger for the smaller DINOv2-S embeddings, see Table 4 in Section D.5. This may suggest that SAEs work well for high dimensional embeddings but struggle with the disentanglement problem as the dimensionality reduces. For sparse probing, the SAE consistently wins, although the difference shrinks for increasing $k_{\mathrm{sp}}$.

## 6. Related Work on Interpreting Transformer Models

The challenge of interpreting large language models and other transformer architectures has prompted significant research. Early efforts focused on probing, where supervised methods are used to determine if predefined concepts have linear representations in model embeddings (Li et al., 2023; Nanda et al., 2023). While effective for verifying known features, probing does not readily uncover novel concepts learned by the model. A prevailing hypothesis for unsupervised concept discovery is that LLM activations represent many distinct features in a superimposed, entangled manner (Elhage et al., 2022). To disentangle these features, dictionary learning approaches aim to find a basis of monosemantic features. SAEs have emerged as a prominent and scalable technique for this task (Bricken et al., 2023; Templeton et al., 2024; Cunningham et al., 2023). Numerous SAE variants have been proposed to enhance performance, including methods like TopK activations to remove the need for $L_1$ penalty tuning (Gao et al., 2024) and Matryoshka representations (Bussmann et al., 2025) to introduce beneficial inductive biases. This rapid development led to the creation of SAEBench (Karvonen et al., 2025), a standardized benchmark for comparing learned dictionaries. The insights gained from disentangled representations are valuable for downstream applications such as understanding circuit-level computations (Lindsey et al., 2025; Ameisen et al., 2025), tracking feature dynamics (Xu et al., 2024), steering model behavior (Zhao et al., 2025), and enabling sparse routing mechanisms (Shi et al., 2025).

## 7. Conclusion

We have shown that alternating optimization algorithms such as KSVD can be scaled to problem sizes relevant for mechanistic interpretability of large transformer models. We achieve competitive results to state-of-the-art SAE approaches evaluated on a variety of standardized metrics for multiple models and modalities. The matching performance

of two different methods may indicate that both methods approach the theoretical limits on these metrics. We have also shown that by focusing on coherence, a property well-studied in the context of sparse dictionary learning, we can explain trends in the autointerpretability and absorption properties. Although DB-KSVD can be scaled to millions of samples, it is not clear whether this is enough to solve the disentanglement problem or whether more scaling is needed. Furthermore, DB-KSVD is algorithmically more complex than SAE approaches, which makes implementation more challenging. Ultimately, we have provided a new perspective on the disentanglement problem and hope that our implementation enables a wide range of applications of the KSVD algorithm.

## Acknowledgements

The authors thank Vikas Sindhwani and Sumeet Singh for valuable discussions and feedback on this work. This work was supported in part by generous funding from Stanford HAI and Google DeepMind.

## Impact Statement

Machine learning models are increasingly used to inform decisions in domains such as healthcare, insurance, and legal and policy contexts. Trusting these systems requires understanding how they reach their conclusions, and which biases and failure modes they exhibit. Mechanistic interpretability studies this question, but the problem remains challenging. Our results suggest that the dictionary learning approach is reaching its practical limits at these problem sizes, and that new approaches will be needed to further understand the internal mechanisms of modern transformer models.

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

# A. Feature Visualization and Autointerpretation Examples

As introduced in Section 1, the superposition hypothesis suggests that transformer embeddings represent many more concepts than their dimensionality would suggest by superimposing multiple monosemantic features within each activation vector. Dictionary learning aims to disentangle these superimposed representations: given an embedding $y$, we find a sparse code $x$ such that $y \approx Dx$, where the columns of $D$ ideally correspond to interpretable, monosemantic features. Figure 3 shows a visualization of this idea using three examples. Given an input image, we display its top-5 most active dictionary features along with their VLM-generated explanations (see Section B.2). For instance, a cocker spaniel image activates features corresponding to various dog-related concepts, demonstrating that the sparse code decomposes the image into its semantic constituents. In this case, the corresponding sparse vector $x$ would have non-zero elements at indices $\{1087, 653, 3189, 3140, 133\}$, and each dictionary element $d_i$ would represent the associated concept. Figure 4 further shows the decoding direction: given a dictionary feature, we display the images that most strongly activate it. Features capture coherent visual concepts, indicating that the learned dictionary elements correspond to interpretable directions in the embedding space.

# B. Experiment Details

## B.1. SAEBench

For the SAEBench benchmark in Section 5, we construct dictionaries using embeddings of the Gemma-2-2B model (Mesnard et al., 2024) evaluated on the Pile Uncopyrighted dataset (Gao et al., 2020). The dictionaries are constructed on a Google Cloud Platform n2-standard-128 machine with 128 simultaneous workers. We learn dictionaries of size $m \in \{4096, 16\,384\}$ with sparsity $k \in \{20, 40, 80\}$ by running 40 iterations of batch size $n = 2^{16}$, thereby using $40 \times 2^{16} = 2\,621\,440$ unique samples. Each sample has dimension $d = 2304$ and is collected as the activations of the 12th layer of the Gemma-2-2B model. Because shuffling the data is difficult, each sample is first stored in a large buffer and then gradually written to a file in a shuffled order.

For the DB-KSVD algorithm, we use regular Matching Pursuit for the sparse encoding which terminates just before the $(k + 1)$th adjacency would be added. During the dictionary update we use an implementation of the Arnoldi Method (Stoppels & Nyman, 2024) with a low tolerance to compute the maximum singular vectors, although we have also observed good results with TSVD (Noack, 2019; Larsen, 1998) or other types of Krylov subspace methods. Arpack (Lehoucq et al., 1998) can be used for validation but does not work well when called from multiple work-

ers. When updating the dictionary elements, the order of their updates is shuffled every time. To initialize the dictionary for DB-KSVD, we sample each dictionary index from a uniform distribution $U(-1/2, 1/2)$ and normalize each column. For Matryoshka DB-KSVD, we use group sizes $\{256, 256, 512, 1024, 2048\}$ for the $m = 4096$ case and group sizes $\{256, 256, 512, 1024, 2048, 4096, 8192\}$ for the $m = 16\,384$ case.

During training, we measure two proxy metrics: mean relative error

$$\frac{1}{n} \sum_i \frac{\|y_i - Dx_i\|_2}{\|y_i\|_2} \tag{6}$$

and variance explained

$$1 - \frac{1}{d} \sum_j \frac{\mathrm{var}((Y - DX)_j)}{\mathrm{var}(Y_j)} \tag{7}$$

where $j$ indexes matrix rows. We plot results for both metrics in Section D.4.

## B.2. DINOv2 Evaluation

We train, evaluate, and compare DB-KSVD and TopK-SAE (Gao et al., 2024) on DINOv2 vision embeddings (Oquab et al., 2024) to validate that our findings extend beyond language models. We use DINOv2-S (ViT-S/14, $d = 384$) and DINOv2-B (ViT-B/14, $d = 768$), extracting CLS token embeddings from the ImageNet-1k training set (approx. $1.3M$ samples). We learn dictionaries of size $m = 4096$ with sparsity $k \in \{16, 32, 64\}$. Unlike for SAEBench, in these experiments we also use matching pursuit to compute the sparse encodings for the SAE to purely compare the quality of the learned dictionary, not the encoding step.

**Evaluation Metrics.** Unlike for language tasks where SAEBench has been proposed as one standardized benchmark for mechanistic interpretability, for the vision domain no such benchmark exists, although ongoing efforts exist (Joseph et al., 2025). For this reason, we propose three metrics that emulate the SAEBench benchmark.

*Variance Explained.* SAEBench uses a "Loss Recovered" metric, a sensible choice for an autoregressive model. However, since vision transformers are typically not trained or evaluated in an autoregressive way, we simply measure the variance explained, a form of reconstruction error, see Eq. (7).

*Sparse Probing.* Following SAEBench k-sparse probing (Karvonen et al., 2025), we evaluate per-class binary classification: for each ImageNet class, we select the $k_{\mathrm{sp}}$ features with highest mean activation difference between that class and all others, train a binary logistic regression on those $k_{\mathrm{sp}}$ features, and report mean accuracy across all 1000 classes.

| Input Image | Top-5 Active Features (by activation magnitude) |
|---|---|
| 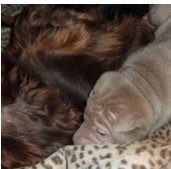 | **Cocker Spaniel**
1. "Chocolate Labrador Retrievers." (#1087)
2. "A group of dogs together." (#653)
3. "Images of relaxed dogs in comfortable, affectionate settings." (#3189)
4. "Black puppies." (#3140)
5. "Scottish Terriers with distinctive fur." (#133) |
| 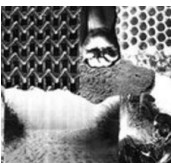 | **Honeycomb**
1. "Whimsical or unconventional objects in everyday settings." (#1509)
2. "Hexagonal patterns in textured surfaces." (#3124)
3. "Honeycomb structures." (#650)
4. "Webpages with text-heavy content and diverse layout designs." (#2263)
5. "Stage curtains in shades of red." (#1174) |
| 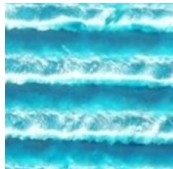 | **Velvet**
1. "Horizontal slatted structures." (#2614)
2. "Vibrant abstract patterns." (#996)
3. "Mop heads and cleaning accessories." (#1763)
4. "Textiles with striped or patterned designs." (#2945)
5. "Long, cylindrical food items." (#2863) |

*Figure 3.* **Sparse decomposition of individual images.** Each row shows random input image (left) with its top-5 most active dictionary features and their VLM-generated explanations (right). The sparse code ($k{=}32$) represents each image as a weighted combination of interpretable concepts. For instance, the cocker spaniel activates features related to dogs, while the honeycomb activates features for hexagonal patterns and structures.

| Feature Explanation | Top-5 Activating Images |
|---|---|
| "Ferrets with close-up facial details." (#273) | |
| "French horns." (#917) | |
| "Carved Halloween pumpkins with glowing faces." (#2923) | |
| "Elephants in various natural and human contexts." (#3274) | |

*Figure 4.* **Feature interpretability via automated interpretability.** Each row shows a learned dictionary feature with its VLM-generated explanation (left) and the top-5 images with highest activation magnitude for that feature (right). The features capture coherent visual concepts ranging from specific animals to objects and patterns, demonstrating that the sparse decomposition learns semantically meaningful directions.

We report results for $k_{\mathrm{sp}} \in \{1, 2, 5\}$ features per class, with $k_{\mathrm{sp}} = 1$ as the primary metric (single feature per concept).

*Autointerpretability.* Autointerpretability for vision models is not as established as for language models, and thus we modify the autointerpretability framework for language models by Bills et al. (2023) to be suitable for a vision task. For each dictionary atom, we select the 8 images with highest activation magnitude and prompt GPT-4o-mini with the prompt below to generate a natural language explanation (some of which are presented in Section A). We then score on a held-out set of 100 images: 50 with the next-highest activations and 50 randomly sampled from the entire dataset. We compute the Spearman correlation between CLIP similarity (explanation, image) and activation values on this scoring set. We report the mean correlation over 100 top-variance features.

> PROMPT: *These images all strongly activate the same feature in a neural network. What visual concept, pattern, or attribute do they share? Be specific but concise. Respond with a single phrase or short sentence describing what this feature detects. Examples: 'dogs playing outdoors', 'red circular objects', 'images with strong diagonal lines', 'close-up faces'.*

**Sign-flip handling.** Unlike for SAEs, DB-KSVD does not require sparse codes to be positive. Thus the sparse coding step produces ∼50% negative coefficients for DB-KSVD (vs ∼5% for SAE). When selecting the highest activation magnitude for autointerpretability, for features where the mean non-zero activation is negative, we first flip the sign of the code and dictionary element before selecting top-activating images. Note that the top-$k$ selection considers only one sign direction per feature, so the reported autointerpretability scores are not artificially inflated by the bidirectional codes.

Enforcing non-negativity throughout DB-KSVD is non-trivial: while a positive variant of Matching Pursuit is straightforward, the SVD-based dictionary update step has no closed-form solution under a positivity constraint on the residual, and modifying it would alter both the convergence behavior and the correctness of the algorithm. Whether the superposition hypothesis truly requires non-negative codes, or whether bidirectional codes can be equally interpretable, remains an open question; Zhu et al. (2025) report comparable SAEBench scores for an SAE variant with bidirectional features, although they do not evaluate autointerpretability.

## C. Computational Complexity and Memory Requirements of DB-KSVD

Each iteration of DB-KSVD requires a sparse encoding step and a dictionary update step.

**Sparse Encoding.** The complexity is dominated by precomputing $D^\top D$ and $D^\top Y$, requiring $O(dm^2)$ and $O(dmn)$ operations respectively. Once precomputed, each of the $k$ iterations per sample reduces to finding the argmax over a column of $D^\top Y$ and updating a cached vector, both $O(m)$ operations. Since $k \ll m \ll n$, the $O(kmn)$ iteration cost is negligible compared to the $O(dm(m + n))$ precomputation.

**Dictionary Update.** Each dictionary update computes the largest eigenvector of $E_{\Omega_j} E_{\Omega_j}^\top \in \mathbb{R}^{d \times d}$. The number of columns in $E_{\Omega_j}$ equals the number of nonzeros in row $j$ of $X$. Assuming nonzeros are evenly distributed across $X$, the density is $k/m$, yielding approximately $kn/m$ nonzeros per row. In practice, this approximation may not always hold and larger factors can occur, but we find it a useful model. Forming $E_{\Omega_j} E_{\Omega_j}^\top$ thus requires $O(d^2 kn/m)$ operations per element, and the Lanczos iterations contribute $O(d^2)$ for a constant number of iterations. Across all $m$ elements, this yields $O(d^2(kn + m))$.

Alternatively, the eigenvalue computation can be done implicitly by computing $E_{\Omega_j}(E_{\Omega_j}^\top v)$ at each Lanczos iteration, avoiding the matrix formation and yielding complexity $O(dkn)$. However, we have not found this beneficial in practice and hypothesize that highly optimized matrix-matrix multiplications outperform a series of matrix-vector products for our problem sizes.

With outer batching (batch size $n_b$), the overall complexity per iteration is then

$$O\left(dm(m + n_b) + d^2(kn_b + m)\right). \qquad (8)$$

**Memory Requirements.** We optimized our implementation to reduce memory requirements and memory movement. The memory requirement of our algorithm is made up of preallocated memory once per program and once per thread. The once-per-program allocations require $O(d(m + n_b))$ bytes for the dictionary buffer and data batch. Additionally, sparse encoding requires $O(m^2 + mn_b)$ for the precomputed $D^\top D$ and $D^\top Y$ matrices. The per-thread allocations require $O(dn_b + d^2)$ bytes for the error matrix $E_{\Omega_j}$ buffer and the $E_{\Omega_j} E_{\Omega_j}^\top$ matrix used in the eigenvalue computation. For our experiments using 128 threads and single-precision floating-point arithmetic, the algorithm requires approximately 80 GB of RAM.

For GPU acceleration, only some parts of the computation are offloaded, requiring $O(mn_b + d^2)$ of GPU memory. If part of the dictionary update is also offloaded to the GPU for

|  |  | CPU | CPU+GPU | Baseline (est.) |
|---|---|---|---|---|
| $m = 4096$ | $k = 20$ | 1428 | 185 | $1.0 \times 10^6$ |
|  | $k = 40$ | 2797 | 292 | $1.6 \times 10^6$ |
|  | $k = 80$ | 5495 | 536 | $2.7 \times 10^6$ |
| $m = 16384$ | $k = 20$ | 433 | 313 | $1.2 \times 10^6$ |
|  | $k = 40$ | 2779 | 649 | $2.1 \times 10^6$ |
|  | $k = 80$ | 6156 | 1243 | $4.2 \times 10^6$ |

*Table 3.* Runtime in seconds for a single batch of $n = 2^{20} = 1\,048\,576$ samples of the Gemma-2-2B embeddings. Baseline results are estimated based on timing results from a smaller problem.

each CPU thread, an additional $O(n_{\text{threads}} \cdot (dkn/m + d^2))$ bytes of GPU memory is required to store $E_{\Omega_j}$ and the resulting matrix product. In practice, depending on the experiment configuration, we required 16 GB to 32 GB of GPU memory with single-precision arithmetic.

# D. Additional Results

## D.1. Additional Timing Results

In Table 3, we present runtime results for one DB-KSVD iteration on a very large batch size of $n = 2^{20}$ samples, complementary to the timing results with a batch size of $n = 2^{16}$ samples presented in Section 5.3. Besides the batch size, the results were obtained in a similar fashion to Section 5.3, except that only 48 workers were used simultaneously on the CPU+GPU machine to limit GPU memory use.

Although in Section 5.3 the introduction of hardware acceleration did not benefit the runtime results, for a large batch size the CPU+GPU approach outperforms the CPU-only approach by a factor of up to 10 for $m = 4096$ and up to 5 for $m = 16\,384$. In practical terms, computing dictionaries with 40 iterations using this batch size and the CPU+GPU approach would take approximately 3 h 15 min for $m = 4096$ or 7 h 15 min for $m = 16\,384$.

## D.2. SAEBench and Coherence Results for 16 384 Dictionary Elements

Figure 5 shows the performance of our trained dictionaries on the SAEBench benchmark similar to Section 5.1 but for $m = 16\,384$ dictionary elements. We observe similar trends as before for the loss recovered, sparse probing, spurious correlation removal and RAVEL metrics. However, unlike for $m = 4096$, the autointerpretability performance of Matryoshka DB-KSVD does not significantly improve compared to DB-KSVD in this case. Additionally, the absorption metric of both methods is much improved relative to almost all SAE-based results.

Figure 6 shows the coherence metric of our dictionaries and comparable SAE-based dictionaries similar to Section 5.2 for $m = 16\,384$. The results look similar to Fig. 2 except that the DB-KSVD results are relatively incoherent even without the Matryoshka modification.

## D.3. SAEBench Pythia-160M Results

Figure 7 shows the performance of our trained dictionaries on the SAEBench benchmark similar to Section 5.1 but using embeddings from the Pythia-160M model. We were unable to find data files for the SAEBench baselines for the RAVEL metric with this model, so do not show comparisons for this metric. For the other metrics, we observe similar trends to the results we get using the Gemma-2-2B embeddings. These results indicate that our conclusions about the performance of DB-KSVD extend beyond a single model.

## D.4. Proxy Metrics

Figure 8 and Fig. 9 display the mean relative error (Eq. (6)) and variance explained (Eq. (7)) proxy metrics that we use to assess model performance during training. At each step, both metrics are computed for the current training batch and a fixed and held out validation batch. We can see that both metrics have mostly flattened out after 40 iterations, which motivates our training setup discussed in Section 5.

Figure 8 and Fig. 9 also provide an empirical indication of the sample requirements for DB-KSVD. Each iteration on the horizontal axis represents an additional $2^{16}$ samples. For $d = 2304$ and both $m = 4096$ and $m = 16\,384$, and irrespective of the sparsity level $k$, we observe that approximately $n = 20 \times 2^{16} \approx 1.3 \times 10^6$ samples are required for convergence. We note that this is not a rigorous sample complexity analysis, but rather an empirical observation of the data requirements for our specific problem setting. A more detailed sample complexity analysis and comparison to SAE methods remains an interesting direction for future work.

## D.5. DINOv2 Results

### D.5.1. FULL RESULTS

In Table 4 we present the full results of the experiments on DINOv2-S and DINOv2-B introduced in Section 5.4. Overall we find a consistent trend of DB-KSVD outperforming SAEs on the variance explained and autointerpretability metrics, and the opposite finding for the sparse probing metric. Notably, the autointerpretability gap significantly widens between SAEs and DB-KSVD for the smaller embedding dimension of DINOv2-S, suggesting that the SAE approach may be less suited to lower dimensional problems. We further note DB-KSVD's remarkable consistency in the autointerpretability scores, in particular for DINOv2-B. Upon

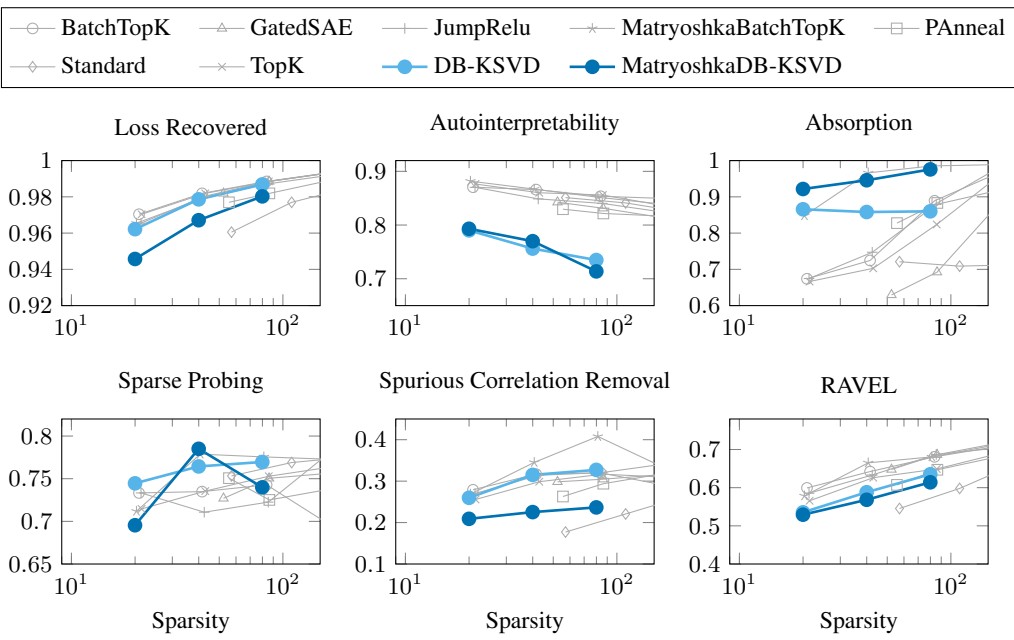

*Figure 5.* Results (higher is better) of our DB-KSVD algorithm and Matryoshka adaptation on the SAEBench benchmark applied to Gemma-2-2B embeddings with 16 384 dictionary elements.

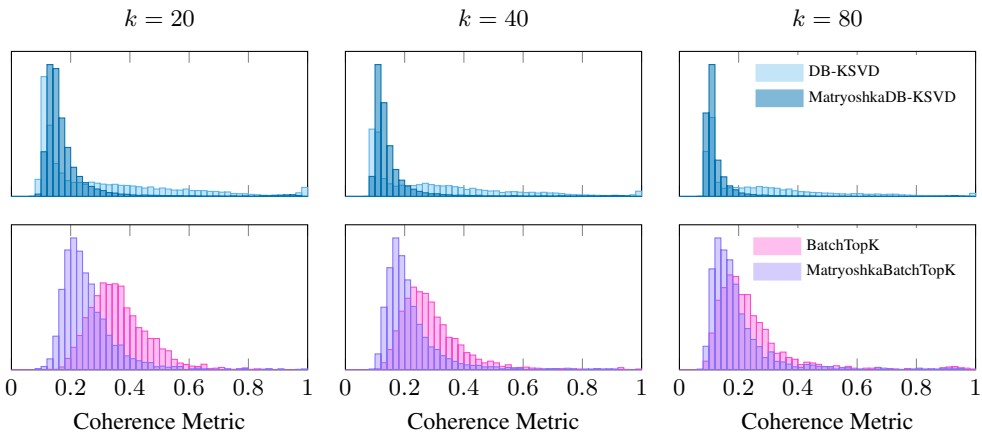

*Figure 6.* Histograms of element-wise coherence metrics for different sparsities. We hypothesize that lower is better. The dictionaries are constructed using the Gemma-2-2B embeddings and comprise $m = 16\,384$ dictionary elements.

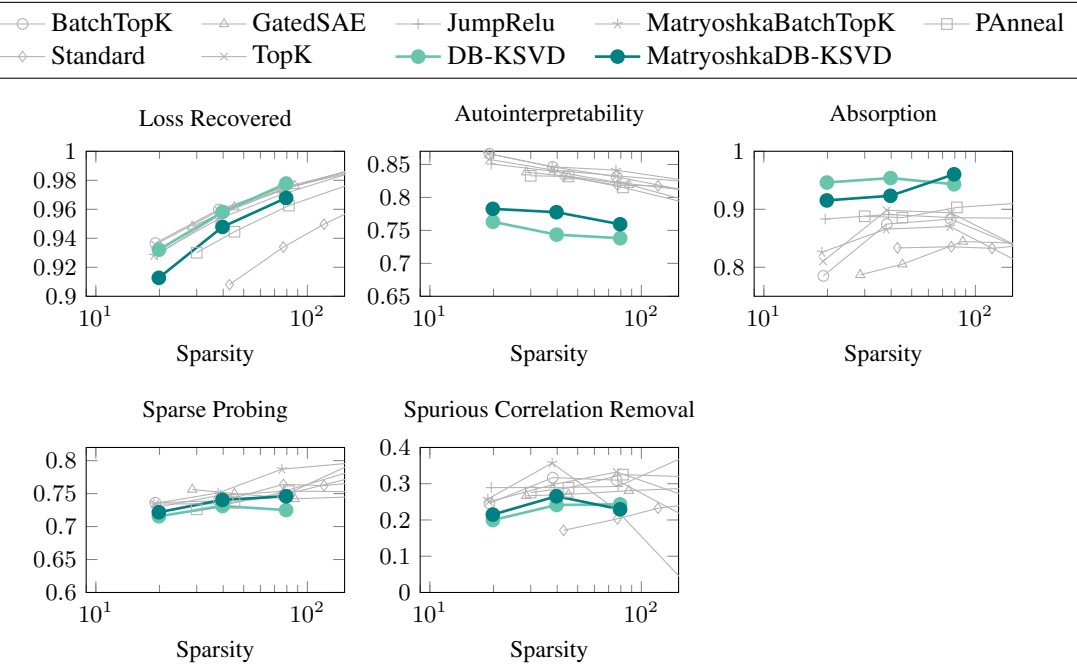

*Figure 7.* Results (higher is better) of our DB-KSVD algorithm and Matryoshka adaptation on the SAEBench benchmark with 4096 dictionary elements on Pythia-160M embeddings.

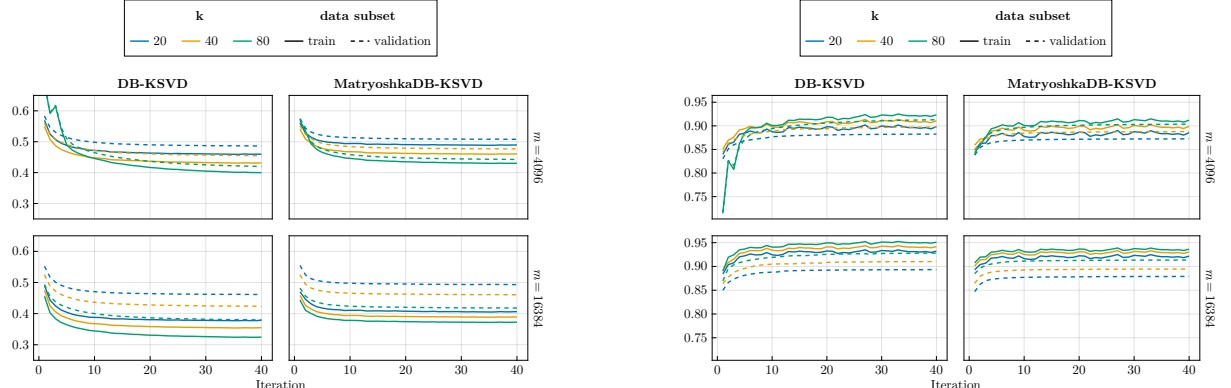

*Figure 8.* Mean relative error (Eq. (6)) for SAEBench training (Section 5) on the Gemma-2-2B model plotted against iterations.

*Figure 9.* Variance explained (Eq. (7)) for SAEBench training (Section 5) on the Gemma-2-2B model plotted against iterations.

inspection, the explanations seem to be "topping out" for DINOv2's dominant visual features, producing very similar image groups and explanations for each $k$.

### D.5.2. EXTENDED SPARSE PROBING RESULTS

Table 5 shows sparse probing accuracy across different numbers of features per class ($k_{sp} \in \{1, 2, 5\}$). SAE consistently outperforms DB-KSVD, with the largest gap at $k_{sp} = 1$ (single feature per concept) and convergence as $k_{sp}$ increases. This suggests SAE features are better at isolating individual concepts, while the gap narrows when multiple features can jointly represent each class.

| Model | $k$ | Var. Explained | | Sparse Probing | | Autointerpretability | |
|---|---|---|---|---|---|---|---|
| | | SAE | DB-KSVD | SAE | DB-KSVD | SAE | DB-KSVD |
| DINOv2-S | 16 | 0.749 | **0.801** | **0.907** | 0.833 | 0.781 | **0.796** |
| | 32 | 0.812 | **0.875** | **0.915** | 0.851 | 0.723 | **0.817** |
| | 64 | 0.866 | **0.942** | **0.924** | 0.861 | 0.694 | **0.803** |
| DINOv2-B | 16 | 0.747 | **0.765** | **0.924** | 0.846 | 0.794 | **0.804** |
| | 32 | 0.794 | **0.817** | **0.932** | 0.893 | 0.797 | **0.802** |
| | 64 | 0.833 | **0.874** | **0.942** | 0.926 | 0.778 | **0.803** |

*Table 4.* Full DINOv2 evaluation with k-sparse probing ($k_{\mathrm{sp}} = 1$). DB-KSVD consistently outperforms SAE on variance explained and autointerpretability; SAE outperforms DB-KSVD on sparse probing.

| Model | $k$ | $k_{\mathrm{sp}} = 1$ | | $k_{\mathrm{sp}} = 2$ | | $k_{\mathrm{sp}} = 5$ | |
|---|---|---|---|---|---|---|---|
| | | SAE | DB-KSVD | SAE | DB-KSVD | SAE | DB-KSVD |
| DINOv2-S | 16 | **0.907** | 0.833 | **0.940** | 0.901 | **0.961** | 0.948 |
| | 32 | **0.915** | 0.851 | **0.943** | 0.908 | **0.963** | 0.952 |
| | 64 | **0.924** | 0.861 | **0.947** | 0.911 | **0.963** | 0.952 |
| DINOv2-B | 16 | **0.924** | 0.846 | **0.954** | 0.927 | **0.972** | 0.965 |
| | 32 | **0.932** | 0.893 | **0.957** | 0.942 | **0.972** | 0.969 |
| | 64 | **0.942** | 0.926 | **0.962** | 0.951 | **0.974** | 0.970 |

*Table 5.* Sparse probing accuracy for varying numbers of features per class $k_{\mathrm{sp}}$. SAE outperforms DB-KSVD across all settings, with the gap largest at $k_{\mathrm{sp}} = 1$.

