# OpenReview forum: "DB-KSVD: Scalable Alternating Optimization for Disentangling High-Dimensional Embedding Spaces"
_ICML.cc/2026/Conference — ICML 2026 regular_

### Official Review · Reviewer_AQwu · 2026-03-02

**Soundness:** 3
**Presentation:** 3
**Significance:** 3
**Originality:** 3
**Overall Recommendation:** 4
**Confidence:** 1

**Summary:**

This paper explores mechanical interpretability by scaling classical dictionary learning to modern Transformer sizes.  The proposed DB-KSVD employs "double-batching" and hardware acceleration to decompose high-dimensional embeddings into sparse, monosemantic features. By matching the performance of SAE, the paper provides a vital alternative to the current SAE-dominated paradigm and proves that traditional alternating optimization can handle millions of samples.

**Compliance With Llm Reviewing Policy:**

Affirmed.

**Final Justification:**

I appreciate the authors’ efforts in providing a rebuttal; however, their response fails to directly resolve the questions I previously raised. I remain of the opinion that there exist significant theoretical limitations and substantial overhead concerns in the work. Therefore, I will retain my current score.

**Key Questions For Authors:**

- Given that DB-KSVD is not mathematically equivalent to the original KSVD, could the authors provide a stability analysis of the algorithm’s convergence toward similar dictionary directions under different initialization conditions?
- We suggest that the authors provide a more detailed comparison of the total power consumption or computational cost between DB-KSVD and Top-K SAE when achieving the same level of Loss Recovered, in order to highlight the advantages of its hardware acceleration.

**Limitations:**

yes

**Strengths And Weaknesses:**

### Strength:

The proposed DB-KSVD features remarkable engineering scalability, achieving a 10,000x speedup over standard KSVD by leveraging Lanczos iterations and optimized matrix precomputations to handle millions of samples. It introduces methodological diversity to the mechanistic interpretability field, demonstrating that traditional alternating optimization can match the performance of SAEs.

### Weakness:

- SAEs are popular because they are simple to code (just gradient descent). DB-KSVD is a lot more complex to implement correctly, which might make it harder for the community to pick up.
- Most of the theoretical section is a recap of existing dictionary learning math. Since the batching approach isn't mathematically identical to the original KSVD, additional rigorous analysis concerning its formal convergence guarantees would have strengthened the theoretical contribution.

---

> ### Author Rebuttal · Authors · 2026-03-29
>
> We thank the reviewer for their feedback and provide two comments.
>
> > [C]ould the authors provide stability analysis of the algorithm’s convergence toward similar dictionary directions under different initialization conditions?
>
> Analyzing the convergence properties of (DB-)KSVD  is a challenging matter and beyond the scope of this paper. A number of papers have been dedicated to this topic, including [1-5], and key questions still remain. Nonetheless, during our experiments we empirically compared the batched approach (DB-KSVD) to the unbatched approach (KSVD) and did not find significant differences in the solutions for large problems, in particular when the order of dictionary updates (inner batch) is shuffled for every outer batch.
>
> > We suggest that the authors provide a more detailed comparison of the total power consumption or computational cost between DB-KSVD and Top-K SAE [...] in order to highlight the advantages of its hardware acceleration.
>
> We believe there may be a misunderstanding here: We do not claim that DB-KSVD outperforms Top-K SAE in terms of runtime or power consumption. Indeed, especially for larger problem sizes, a fully GPU-accelerated SAE implementation likely outperforms DB-KSVD (CPU-only or CPU+GPU) in terms of runtime, and likely also in terms of power consumption. Instead, we provide a significant speedup for the KSVD algorithm itself. Table 1 and 3 report runtime requirements for the different KSVD implementations, and Appendix C includes a complexity analysis. The runtime requirements are closely related to power consumption.
>
> [1]: M. Aharon, M. Elad, and A. Bruckstein, "K-SVD: An Algorithm for Designing Overcomplete Dictionaries for Sparse Representation," IEEE Transactions on Signal Processing, 2006.
>
> [2]: C. Bao, H. Ji, Y. Quan, and Z. Shen, "Dictionary Learning for Sparse Coding: Algorithms and Convergence Analysis," IEEE Transactions on Pattern Analysis and Machine Intelligence, 2015.
>
> [3]: A. Agarwal, A. Anandkumar, P. Jain, and P. Netrapalli, "Learning Sparsely Used Overcomplete Dictionaries via Alternating Minimization," SIAM Journal on Optimization, 2016.
>
> [4]: S. Ruetz and K. Schnass, "Convergence Regions of Alternating Minimization Algorithms for Dictionary Learning," SIAM Journal on Optimization, 2026.
>
> [5]: N. S. Chatterji and P. L. Bartlett, "Alternating Minimization for Dictionary Learning: Local Convergence Guarantees," arXiv:1711.03634, 2019.

---

> > ### Author Rebuttal · Reviewer_AQwu · 2026-04-01
> >
> > The author's response did not directly address my question. I still believe there are theoretical and overhead limitations, so I will keep my score.

---

### Official Review · Reviewer_6oPt · 2026-03-05

**Soundness:** 3
**Presentation:** 3
**Significance:** 3
**Originality:** 2
**Overall Recommendation:** 4
**Confidence:** 3

**Summary:**

This paper proposes Double-Batch KSVD (DB-KSVD), a scalable adaptation of the classic KSVD dictionary learning algorithm, and applies it to the problem of disentangling transformer embeddings for mechanistic interpretability. The key engineering contributions include parallel Matching Pursuit with precomputed Gram matrices, batched dictionary updates distributable across CPU workers, outer mini-batching for out-of-core problem sizes, and optional GPU offloading — collectively yielding up to 10,000x speedup over a baseline KSVD implementation. The paper also adopts a Matryoshka structuring technique (from the SAE literature) to encourage dictionary incoherence. DB-KSVD is evaluated on Gemma-2-2B and Pythia-160M language model embeddings using the six-metric SAEBench benchmark, and on DINOv2-S/B vision model embeddings using custom-adapted metrics. The authors find that DB-KSVD achieves competitive performance with several established SAE variants, and argue this suggests both optimization families may be approaching the practical limits of dictionary learning for these problem sizes.

**Compliance With Llm Reviewing Policy:**

Affirmed.

**Key Questions For Authors:**

1. **Non-negativity constraint on sparse codes.** What happens if you enforce non-negative Matching Pursuit (selecting only positive inner products at each greedy step)? This would make DB-KSVD's encoding more comparable to SAEs' ReLU-based encoding and could address the ~50% negative coefficients. If this improves autointerpretability and coherence, it would significantly strengthen the paper; if it does not, that itself would be an informative result about the role of non-negativity in dictionary learning for interpretability.

2. **Statistical significance of performance differences.** Can you provide confidence intervals (e.g., via bootstrapping over evaluation samples or multiple random seeds) for the SAEBench metrics? Specifically, is the autointerpretability gap between DB-KSVD and the best SAE variants statistically significant? If the differences fall within noise, the "competitive" claim is strongly supported; if they are significant, the narrative should be adjusted.

3. **Sensitivity to sparse encoding algorithm.** You chose Matching Pursuit for computational efficiency over OMP or LASSO. Have you measured how much solution quality degrades relative to these alternatives (even on a subsample)? If a more accurate sparse encoding method substantially closes the interpretability gap at modest computational cost, it would change the assessment of whether the current results reflect DB-KSVD's potential or its ceiling.

**Limitations:**

Yes — the authors discuss several relevant limitations: DB-KSVD's algorithmic complexity relative to SAEs, uncertainty about whether current scale is sufficient for full disentanglement, and the coherence issue for learned dictionaries. The honest reporting of underperformance on autointerpretability and the tradeoffs of Matryoshka structuring is appreciated. Two additional limitations that would benefit from discussion are: (i) the practical resource requirements (~80 GB RAM, 128 threads) and how they compare to SAE training costs, and (ii) the implications of pervasive negative coefficients for the interpretability of learned features.

**Strengths And Weaknesses:**

### Strengths:

S1. Well-motivated algorithmic contribution with substantial practical impact. Scaling KSVD from hundreds of dimensions and tens of thousands of samples to thousands of dimensions and millions of samples is a meaningful engineering achievement. The specific modifications (precomputed D^T D and D^T Y, inner batching with shuffled worker synchronization, outer mini-batching, and selective GPU offloading) are clearly described and yield a demonstrated ~10,000x speedup that reduces training time from estimated weeks to minutes (Section 5.3, Table 1). The computational complexity analysis in Appendix C is thorough.

S2. Comprehensive evaluation on a standardized benchmark. Using SAEBench with six metrics across multiple models (Gemma-2-2B, Pythia-160M) and against seven SAE variants provides a rigorous, apples-to-apples comparison. The SAE baselines come from independently tuned, publicly available dictionaries (Karvonen et al., 2025), which significantly mitigates strawman concerns.

S3. Valuable cross-community insight. The finding that an entirely different optimization approach (alternating optimization) reaches comparable performance to SAEs is scientifically interesting. It bridges the classical signal processing / sparse coding literature with the modern mechanistic interpretability community, and the observation that two fundamentally different methods converge to similar solution quality is a useful data point.

S4. Multi-modality validation. Extending experiments to DINOv2-S and DINOv2-B vision embeddings, including the observation that DB-KSVD's advantage over SAEs widens for lower-dimensional embeddings (Table 4, DINOv2-S), adds meaningful breadth and suggests the findings are not idiosyncratic to a single domain.

### Weaknesses:

W1. No confidence intervals or statistical significance analysis. All results are reported as point estimates without error bars, standard deviations, or significance tests. Given that the central claim is "competitive" performance — i.e., that differences between DB-KSVD and SAEs are small — it is essential to establish whether these differences are statistically meaningful or within noise. For example, in Table 2, the sparse probing gap between SAE (0.932) and DB-KSVD (0.893) at ksp=1 could be significant or could reflect variance from the evaluation pipeline. Without this information, the reader cannot assess whether DB-KSVD truly matches SAE performance or is systematically worse on certain metrics.

W2. The autointerpretability gap is inadequately explained. DB-KSVD consistently underperforms on autointerpretability relative to most SAE variants across both dictionary sizes and both language models (Figures 1, 5, 7). For a paper motivated by mechanistic interpretability, this is arguably the most important metric. The paper hypothesizes a link to dictionary coherence (Section 5.2) but provides only correlational evidence. It is also unclear why Matryoshka DB-KSVD closes the gap for m=4096 but not for m=16384. A more rigorous investigation — e.g., ablations that directly manipulate coherence, or analysis of what the less-interpretable features look like — would strengthen the paper.

W3. The ~50% negative coefficient issue is underexplored. DB-KSVD produces approximately 50% negative sparse codes (Section B.2), compared to ~5% for SAEs. This is a fundamental asymmetry that the paper handles via post-hoc sign flipping. Does this indicate that DB-KSVD is fitting a different generative model? Does it conflict with the superposition hypothesis, which assumes non-negative feature activations? Would imposing a non-negativity constraint during Matching Pursuit change the results? This deserves deeper treatment, as it may explain both the coherence and autointerpretability gaps.

W4. Limited novelty in individual algorithmic components. Each modification is individually well-known: precomputing Gram matrices for Matching Pursuit (Davis et al., 1997), batched dictionary updates, mini-batching, GPU offloading. The theoretical section reviews known identifiability results without contributing new theory. The Matryoshka structuring is adopted from Bussmann et al. (2025). While the combination at scale is valuable, the paper would benefit from being more explicit about which components are novel versus known.

---

> ### Author Rebuttal · Authors · 2026-03-29
>
> We thank the reviewer for their thoughtful feedback and provide responses to the key questions below.
>
> > What happens if you enforce non-negative Matching Pursuit [...]?
>
> Enforcing strictly positive codes in (DB-)KSVD would be a significant change to the algorithm, with poorly understood implications on convergence and correctness. While the matching pursuit stage is easy to modify, the dictionary update relies on computing the singular-value decomposition (SVD) on the residual where positivity can not easily be enforced. Conversely, training DB-KSVD the regular way but enforcing positive codes only at test time violates the algorithm's assumptions and may undermine its performance.
>
> Notwithstanding, we point out that the autointerpretability analysis does select the “top-k” codes (i.e., no absolute value operator). In other words, only features “in one direction” are grouped and interpreted. We therefore believe that the results are representative of the true performance as is.
>
> It remains an interesting research question if the superposition hypothesis is correctly stated as requiring only positive coefficients. We are not aware of work thoroughly ablating the choice of positive code enforcement for autointerpretability. In one related work [1] the authors study several SAEBench metrics for an SAE without positive code enforcement. The authors find a minor performance increase in the metrics, although notably autointerpretability is not analyzed. We look forward to further research exploring the assumptions of the superposition hypothesis and implications for autointerpretability.
>
> [1]: X. Zhu, M. M. Khalili, and Z. Zhu, "AbsTopK: Rethinking Sparse Autoencoders For Bidirectional Features," arXiv:2510.00404, 2025.
>
> > Is the autointerpretability gap between DB-KSVD and the best SAE variants statistically significant?
>
> Although we can not offer full confidence intervals, in this work we show consistent trends across different models and problem sizes. Our narrative is in line with these trends: DB-KSVD performs competitively on most metrics but concedes some performance gap for autointerpretability when compared to SAE-based approaches. The autointerpretability gap can be understood from the perspective of coherence, and Matryoshka structuring jointly improves incoherence and autointerpretability in all cases; a trend that is consistent with the SAE literature.
>
> > Have you measured how much solution quality degrades relative to [OMP or LASSO]?
>
> As part of our experiments we also implemented OMP and ran it on a reduced problem size. We did not find significant performance improvements despite drastically increased runtime. We hypothesize that due to the high dimensionality most dictionary elements are “almost orthogonal” to each other in the first place. Therefore, convergence barely benefits from the additional orthogonalization. We note that OMP increases the number of computations by at least a factor of $k$ (the sparsity), making it prohibitively expensive for the full problem.
>
> Regarding LASSO and similar gradient-based methods, we highlight that this paper specifically focuses on an alternative to gradient-based methods, namely Alternating Optimization via KSVD. We consider gradient-based encoders such as LASSO or learned encoders to be out of scope for this paper.

---

> > ### Author Rebuttal · Reviewer_6oPt · 2026-03-31
> >
> > I thank the authors for their responses. The OMP result (Q3) is informative and I appreciate the effort — it strengthens the case that Matching Pursuit is a reasonable choice for this setting.
> >
> > However, the two most critical concerns remain unaddressed:
> >
> > - **W1 (error bars):** Confidence intervals are still not provided. The authors point to "consistent trends," but the central claim of the paper is that DB-KSVD is *competitive* with SAEs — a claim about small differences being insignificant, which inherently requires statistical evidence. Without it, this claim remains unsubstantiated.
> > - **W3 (negative coefficients) / Q1 (non-negativity):** I understand the algorithmic difficulty, but this does not resolve the scientific question of whether the ~50% negative codes are a fundamental limitation explaining the autointerpretability gap. The cited AbsTopK work (Zhu et al., 2025) did not analyze autointerpretability, so it does not help here.
> >
> > The response to Q2 (autointerpretability gap) restates the paper's existing discussion without new evidence.
> >
> > I maintain my score of **4**. The engineering contribution is solid, but the missing statistical analysis and the unexplored negative-coefficient issue prevent a stronger recommendation.

---

### Official Review · Reviewer_eF3r · 2026-03-13

**Soundness:** 3
**Presentation:** 3
**Significance:** 3
**Originality:** 3
**Overall Recommendation:** 5
**Confidence:** 4

**Summary:**

This paper proposes DB-KSVD, a scalable dictionary learning algorithm applied to disentangle high-dimensional features learned by large-scale (transformer) models. By modernizing the classic KSVD algorithm and adapting it to better utilize modern hardware, DB-KSVD can scale to much larger dimensions and number of samples. Experiments on language and vision settings reveal that the proposed approach achieves competitive results compared to SAE-based methods that are more widely used. This work shows that classical methods can be feasibly scaled and applied to modern learning problems and leave ample room for future research.

**Compliance With Llm Reviewing Policy:**

Affirmed.

**Key Questions For Authors:**

Please see the weaknesses.

**Limitations:**

Yes.

**Strengths And Weaknesses:**

Strengths:
1. This paper proposes a solid and technical adaptation of the classical KSVD algorithm so that it can be scaled to much larger problem space. It also proposes a variant which incorporates Matryoshka structure, further improving performance in certain metrics.
2. Experiments are conducted on both language and vision settings, showcasing the flexibility and broad applicability of the proposed approach. Overall, the experiment protocol is well-designed and well-executed.
3. The results also serve as evidence that SAEs are indeed effective approaches for mechanistic interpretability.
4. The paper is overall well written and presented.

Weaknesses:
1. The fact that DB-KSVD performs similarly compared to SAEs, which typically are much easier to train and better utilize modern hardware. This may limit incentive for practitioners to adopt the proposed approach.
2. While I like the writing, I find the authors spend more space on the preliminaries of dictionary learning. I would suggest the authors instead present a more formal exposition of the classic KSVD algorithm.

---

> ### Author Rebuttal · Authors · 2026-03-29
>
> We thank the reviewer for their feedback. We acknowledge in the paper that DB-KSVD comes with added implementation challenges compared to purely gradient-descent based algorithms. Gradient-descent based algorithms, such as SAE, are both algorithmically simpler and can rely on the extensive software and hardware foundations developed in the recent decade.
> Nonetheless, we note that as part of this publication we make our implementation of DB-KSVD available, including an accessible interface for Python, Julia, and R. We hope that the combination of implementation and analysis in this work provides future researchers with a practical and efficient way to apply KSVD to their needs.

---

> > ### Author Rebuttal · Reviewer_eF3r · 2026-04-04
> >
> > I appreciate the authors for agreeing to open-source the implementation, which is always great for reproducibility and future research. I maintain my recommendation for accepting.

---

### Official Review · Reviewer_HjyE · 2026-03-14

**Soundness:** 3
**Presentation:** 3
**Significance:** 3
**Originality:** 3
**Overall Recommendation:** 4
**Confidence:** 2

**Summary:**

The paper proposes DB-KSVD, a scalable adaptation of the classic KSVD dictionary learning algorithm, designed to disentangle transformer embeddings for mechanistic interpretability. The key engineering contribution is making KSVD work at the scale required for modern transformers (millions of samples, thousands of dimensions) through parallel matching pursuit, batched dictionary updates, and optional GPU offloading. The authors evaluate on SAEBench using Gemma2-2B and Pythia-160M embeddings, and additionally on DINOv2 vision embeddings, showing competitive performance with SAE-based approaches. This paper also includes a useful discussion of theoretical identifiability conditions for dictionary learning and adopts the Matryoshka structuring from recent SAE work.

**Compliance With Llm Reviewing Policy:**

Affirmed.

**Key Questions For Authors:**

n/a

**Strengths And Weaknesses:**

Strengths
1. The core engineering contribution is substantial and well-executed. The 10,000× speedup over baseline KSVD is impressive and clearly explained.
2. The algorithmic modifications are individually well-motivated and the complexity analysis is thorough.
3. The paper asks a genuinely interesting question: are SAEs actually good at solving the dictionary learning problem, or are they just convenient? This question along with the benchmarking provides valuable insights for the interpretability community.
4. The theoretical discussion in Section 3 connecting identifiability, coherence, sparsity, and sample complexity to practical design choices is solid.
5. The evaluation is comprehensive.

Weaknesses

1. MP is greedy and known to produce suboptimal sparse codes compared to OMP, LASSO, or learned encoders. The paper justifies MP purely on computational grounds but never ablates this choice. If DB-KSVD with OMP or basis pursuit substantially outperformed MP-based DB-KSVD, it would change the story. This ablation is missing and important.
2. The paper shows DB-KSVD produces highly coherent dictionaries (fig. 2), especially at low k, with some elements nearly perfectly aligned. Matryoshka structuring helps but doesn't fully resolve it. High coherence directly undermines the identifiability guarantees the paper itself discusses in Section 3.2. This is somewhat circular: the theory says incoherence is needed, the algorithm produces coherent dictionaries, and the paper then hypothesizes this explains the autointerpretability gap, but offers no solution beyond Matryoshka, which has its own tradeoffs (degrades loss recovered and SCR).
3. Missing synthetic validation. As a paper that leans heavily on dictionary learning theory, this work lacks the experiments on synthetic data where ground truth D and X are known is a notable gap. Such experiments would let the authors measure actual recovery quality rather than relying on downstream proxy metrics, and would help disambiguate whether competitive SAEBench scores reflect good dictionary recovery or just that both methods find similarly useful but potentially wrong dictionaries.

---

> ### Author Rebuttal · Authors · 2026-03-29
>
> We thank the reviewer for their thoughtful feedback and offer two thoughts to the discussion.
>
> > MP is greedy and known to produce suboptimal sparse codes compared to OMP, LASSO, or learned encoders. [...]
>
> As part of our experiments we did implement OMP and ran it on a reduced problem size. We did not find significant performance improvements despite drastically increased runtime. We hypothesize that due to the high dimensionality most dictionary elements are “almost orthogonal” to each other in the first place. Therefore, convergence barely benefits from the additional orthogonalization. We note that OMP increases the number of computations by at least a factor of $k$ (the sparsity), making it prohibitively expensive for the full problem.
>
> Regarding LASSO and similar methods, we highlight that this paper is about Alternating Optimization algorithms, notably KSVD. We consider gradient-based encoders such as LASSO or learned encoders to be out of scope for this paper.
>
> > Missing synthetic validation. [...]
>
> As part of our experiments we also implemented and wrote up a basic synthetic experiment to understand DB-KSVD convergence properties. We are happy to add this section to the appendix of the final paper. We point out, however, that thoroughly analyzing the convergence properties of (DB-)KSVD is beyond the scope of this paper. This analysis is a challenging matter, and a wealth of related papers have been dedicated to this topic, including [1-5].
>
> [1]: M. Aharon, M. Elad, and A. Bruckstein, "K-SVD: An Algorithm for Designing Overcomplete Dictionaries for Sparse Representation," IEEE Transactions on Signal Processing, 2006.
>
> [2]: C. Bao, H. Ji, Y. Quan, and Z. Shen, "Dictionary Learning for Sparse Coding: Algorithms and Convergence Analysis," IEEE Transactions on Pattern Analysis and Machine Intelligence, 2015.
>
> [3]: A. Agarwal, A. Anandkumar, P. Jain, and P. Netrapalli, "Learning Sparsely Used Overcomplete Dictionaries via Alternating Minimization," SIAM Journal on Optimization, 2016.
>
> [4]: S. Ruetz and K. Schnass, "Convergence Regions of Alternating Minimization Algorithms for Dictionary Learning," SIAM Journal on Optimization, 2026.
>
> [5]: N. S. Chatterji and P. L. Bartlett, "Alternating Minimization for Dictionary Learning: Local Convergence Guarantees," arXiv:1711.03634, 2019.

---

> > ### Author Rebuttal · Reviewer_HjyE · 2026-04-03
> >
> > i appreciate the authors' rebuttal.

---

> > > ### Author Response · Authors · 2026-04-03
> > >
> > > Thank you for reviewing our rebuttal, we are glad to see that we were able to address the concerns. In this light, we kindly wanted to check whether you feel the current score still reflects your assessment of the paper, or if you may wish to update the score. Either way, we greatly appreciate your time and feedback.

---

### Decision · Program_Chairs · 2026-04-30

**Decision:**

Accept (regular)

**Comment:**

This paper proposes DB-KSVD, a scalable dictionary learning algorithm applied to disentangle high-dimensional features learned by large-scale (transformer) models.  The reviewers agreed that the method is well-motivated and experimental evaluations are comprehensive.  The paper is overall well-written and presented. The authors' rebuttal has addressed the concerns about the ablation studies and missing synthetic validation as well as the open-source implementation.  After rebuttal, there are still some concerns on the missing statistical analysis and the unexplored negative-coefficient issue, limited theoretical analysis, and substantial computational cost, which are not fully resolved.  The authors are strongly encouraged to address or clearly discuss these issues in the final version.